# Ciliated Cells in Ovarian Cancer Decrease with Increasing Tumor Grade and Disease Progression

**DOI:** 10.3390/cells11244009

**Published:** 2022-12-11

**Authors:** Michael T. Richardson, Maria Sol Recouvreux, Beth Y. Karlan, Ann E. Walts, Sandra Orsulic

**Affiliations:** 1Department of Obstetrics and Gynecology, David Geffen School of Medicine, University of California Los Angeles, Los Angeles, CA 90095, USA; 2Jonsson Comprehensive Cancer Center, University of California Los Angeles, Los Angeles, CA 90095, USA; 3Department of Pathology and Laboratory Medicine, Cedars-Sinai Medical Center, Los Angeles, CA 90048, USA; 4VA Greater Los Angeles Healthcare System, Los Angeles, CA 90095, USA

**Keywords:** fallopian tube, ciliated cells, secretory cells, ovarian cancer, serous, mucinous, endometrioid, clear cell, CAPS, FOXJ1

## Abstract

Ciliated cell markers expressed in epithelial ovarian cancers (EOC) are associated with improved survival. We examined the distribution of cells expressing ciliated cell markers in various EOC histologies and stages. Immunohistochemistry and/or multiplex immunofluorescence were used to determine the expression of FOXJ1 and/or CAPS (ciliated cell markers) in tissue microarrays including 4 normal fallopian tubes, 6 normal endometria, 16 cystadenomas, 25 borderline tumors, 21 low-grade carcinomas, and 118 high-grade carcinomas (HGSOC) (46 serous, 29 endometrioid, 30 clear cell, 13 mucinous). CAPS+ cells were observed in normal fallopian tubes and endometria and in ~85% of serous benign and borderline tumors and low-grade carcinomas but only in <40% of HGSOC. mRNA data from an independent cohort showed higher FOXJ1 and CAPS expression in serous borderline tumors and low-grade carcinomas compared to HGSOC. In HGSOC, ciliated cell-positive markers were observed in 52% primary tumors compared to 26% of patient-matched synchronous metastases, and 24% metachronous metastases (*p* = 0.009). mRNA data from an independent HGSOC cohort showed lower levels of CAPS in metastases than in primary tumors (*p* = 0.03). Overall, the study revealed that ciliated cells were less common in mucinous EOC, the percentage of ciliated cell marker-positive cases decreased with increasing grade, and the percentage of ciliated cells decreased in HGSOC metastases compared to patient-matched primary tumors.

## 1. Introduction

In the United States, nearly 20,000 women are diagnosed with ovarian cancer annually and 12,000 succumb to their disease [1]. Epithelial ovarian cancer (EOC) is a heterogeneous disease that includes serous, endometrioid, mucinous and clear cell histologic subtypes. These histologic subtypes are further divided into high-grade, low-grade, and borderline (previously termed low-malignant potential or LMP) tumors [2]. High-grade serous ovarian cancer (HGSOC) which accounts for more than two-thirds of ovarian cancer deaths [3] was initially thought to originate from the ovarian surface epithelium or peritoneum [4,5]. However, more recent studies indicate tumor origin from the secretory cells at the fimbriated end of the fallopian tube where precursor lesions, including serous tubal intraepithelial carcinoma (STIC) and a TP53 mutation signature, are typically identified [3,6,7].

The fallopian tube epithelium is composed primarily of secretory cells and ciliated cells. Whereas the secretory cells produce the liquid film which overlays the tubal epithelium and are more numerous in all segments of the tube [8,9,10], ciliated cells help transport the ovum through the tube and are more numerous in the fimbriated end of the tube, decreasing in number toward the uterus [11]. Ciliated cells are also thought to aid in follicular fluid clearance, thereby possibly reducing the genotoxic stress that follows ovulation [4]. Ciliated cells are also present in the endometrium (both eutopic endometrium and endometriosis) [12] and in some ovarian cortical inclusion cysts (CIC) [13,14,15].

Relatively little is known about the role of ciliated cells in the development of EOC. Ciliated cell dysfunction has been proposed to enable DNA damage and carcinogenesis [16,17], although the mechanism is unknown. The presence of ciliated cell markers has also been shown to be associated with improved survival in EOC [18].

We sought to better understand the relationships between ciliated cells and the biologic behavior of various benign and malignant epithelial ovarian tumors and histologies. We utilized patient tumor tissue microarrays (TMAs), immunohistochemistry (IHC) for ciliated cell marker-positive (FOXJ1 and/or CAPS) cells, multiplex immunofluorescence (mIF) for ciliated cell marker-positive (CAPS, TUBB4) cells and secretory cell marker (PAX8), and mRNA expression profiles to characterize the distribution of ciliated and secretory cell markers in normal fallopian tubes and endometria, benign and borderline ovarian tumors, and various histologic subtypes and grades of EOC. We also utilized a TMA of patient-matched tumors to compare the presence of ciliated cell markers in primary, synchronous metastatic, and metachronous metastatic/recurrent HGSOC.

## 2. Materials and Methods

### 2.1. TMAs

Our study was approved by the UCLA and Cedars-Sinai Medical Center Institutional Review Boards. We utilized commercial and in-house generated TMAs with de-identified formalin-fixed, paraffin-embedded (FFPE) patient tissue samples and associated clinicopathologic data (age, histologic subtype, malignant potential, grade, and stage). Four TMAs with non-overlapping patient tumors were used: TMA1 (OV991 Ovarian Tumor Series; US Biomax, Inc, Derwood, MD, USA; a single 1.2 mm core/case), TMA2 (CJ3 Ovary Cancer; Super Bio Chips; a single 2 mm core/case), TMA3 (CHTN OvCa2 Ovarian Carcinoma Survey; four 0.6 mm cores/case), and TMA4 (Ovarian Cancer Histologic Subtypes; in-house generated TMA; two 1.5 mm cores/case). Appendix A shows all cases that were analyzed after excluding cases with missing cores or insufficient tissue and cases irrelevant to this study.

We also generated a “HGSOC Progression” TMA comprising triplicate 1 mm cores of HGSOC from 42 patients. In this TMA, each patient was represented by 3 tumors: primary tumor, concurrent (synchronous) metastasis, and recurrent (metachronous) metastasis. The primary tumors and concurrent/synchronous metastases were chemotherapy-naive while recurrent/metachronous metastases were obtained following treatment (usually 6 cycles of platinum/taxane combination chemotherapy).

### 2.2. Multiplex Immunofluorescence (mIF)

Opal mIF staining was performed by the UCLA Translational Pathology Core Laboratory based on tyramide signal amplification (TSA). The Opal Polaris 7-Color Automation IHC Kit (Cat#NEL871001KT, Akoya Biosciences, Marlborough, MA, USA) was used with ER2 30 min retrieval and 1 h room temperature incubation. Staining was performed consecutively using each of the following antibodies: Paired Box 8 (PAX8, prediluted rabbit polyclonal antibody 363A-18, Cell Marque, Rocklin, CA, USA), calcyphosine (CAPS, Atlas Antibodies Cat#HPA043520, RRID:AB 10964138, 1:1000 dilution, Sigma-Aldrich, St. Louis, MO, USA), and tubulin beta 4 (TUBB4, Sigma-Aldrich, St. Louis, MO, USA), monoclonal anti-acetylated tubulin T7451, 1:500 dilution), and 4′,6-diamidino-2-phenylindole (DAPI) (Cat# SKU FP1490, Akoya Biosciences, Marlborough, MA, USA). The slides were scanned at 20X with the Vectra Polaris scanner (Akoya Biosciences, Marlborough, MA, USA) and data from the multispectral camera were analyzed using the imaging InForm automated image analysis software (Akoya Biosciences, Marlborough, MA, USA). Images were scored at 10X magnification using Phenochart whole slide viewer (Akoya Biosciences, Marlborough, MA, USA).

### 2.3. Immunohistochemistry (IHC)

IHC was performed by the UCLA Translational Pathology Core Laboratory using the Leica Bond RX processor based on Protocol F using Bond Polymer Refine Detection kit (Cat#DS9800, Leica Biosystems, Nussloch, Germany), ER2 buffer (BOND Epitope Retrieval Solution 2, Leica Biosystems, Cat#AR9640), DakoCytomation Envision System Labelled Polymer HRP anti rabbit (Agilent, Santa Clara, CA, USA, K4003) and antibodies against FOXJ1 (Sigma-Aldrich, St. Louis, MO, USA, Atlas Antibodies Cat#HPA005714, RRID:AB_1078902, 1:100 dilution) and CAPS (Sigma-Aldrich, Atlas Antibodies Cat#HPA043520, RRID:AB_10964138, 1:1000). Hematoxylin was used as the counterstain. The slides were scanned at 40X with the Aperio ScanScope AT (Aperio, Vista, CA, USA).

### 2.4. Scoring and Interpretation

The percentage of CAPS+ cells in different stages of ovarian cancer progression represented in TMAs 1–4 was evaluated by mIF. CAPS, TUBB4, and DAPI channels were used for scoring while the PAX8 channel was turned off to facilitate scoring of CAPS+ cells. Missing cores and cores containing <50 epithelial cells were excluded from analysis. For TMA3 (four adjacent cores per case) and TMA4 (two adjacent cores per case), all cores/case were scored as one contiguous sample. Percentage of CAPS+ cells in all epithelial cells per case was estimated by eye in increments of 10 in 10X Phenochart viewer images. Cases containing >5 CAPS+ cells but <1%of CAPS+ cells per case were scored as 1%. Cases without CAPS+ cells or with <5 CAPS+ cells per case were scored as 0%.

Since HGSOC cases typically contained no or only a few ciliated cell marker-positive cells, which could be missed by either FOXJ1 or CAPS staining, three non-consecutive “HGSOC Progression” TMA sections were stained using IHC for FOXJ1, IHC for CAPS, and mIF for CAPS. A case was considered positive when >5 cells in one or more cores stained positive by at least one of the three staining methods. The three methods yielded similar scores. Re-review of the cores with discrepant results revealed that positive cells had been undercounted by one or two of the three methods because the tissue containing positive cells was folded or damaged in individual TMA sections.

### 2.5. Transcriptome Dataset

The Ovarian Cancer Database of Cancer Science Institute Singapore, CSIOVDB (http://csiovdb.mc.ntu.edu.tw/CSIOVDB.html, accessed on 22 May 2022) is a transcriptomics database of 3431 human ovarian neoplasms, including serous, endometrioid, mucinous, and clear cell carcinomas, and serous-LMP (borderline) tumors, and mucinous-LMP (borderline) tumors [19]. Expression levels of FOXJ1 and CAPS (ciliated cell markers) and PAX8 (a secretory cell marker) were plotted using the CSIOVDB online tool, comparing ovarian tumor histology and malignant potential.

### 2.6. Statistical Analyses

mRNA expression levels, measured in mean and quantiles, were compared using the Mann–Whitney Test. CAPS+ expression between histology was determined with Chi-squared testing. A *p*-value of 0.05 was considered significant.

## 3. Results

### 3.1. Ciliated Cells Are Frequently Present in Ovarian Cystadenomas, Borderline Tumors, and Low-Grade Carcinomas

To validate FOXJ1 and CAPS antibodies as markers of ciliated cells, we assessed their localization by IHC (FOXJ1) and mIF (CAPS) in normal fallopian tubes and proliferative endometria. In ciliated cells from both sites, FOXJ1 was expressed in the nucleus (Figure 1A,C), CAPS was present in all cellular compartments including at the base of cilia, and tubulin highlighted intracellular skeletal structures and cilia (Figure 1B,D).

In ovarian cystadenomas, borderline tumors, and low-grade carcinomas, CAPS+ cells were typically associated with easily recognized cilia (Figure 2A–C) while cilia were rarely recognizable in high-grade carcinomas (Figure 2D–F). In ovarian high-grade carcinomas, even in the absence of distinct cilia, it was clear that CAPS+ cells did not show nuclear staining for PAX8 and thus were immunophenotypically distinct from the PAX8+ carcinoma cells (Figure 3).

While CAPS+ cells were detected in approximately 85% of serous cystadenomas, borderline tumors, and low-grade carcinomas, they were detected in <40% of HGSOCs. The average percentage of CAPS+ cells decreased with increasing tumor grade (40% in benign serous cystadenomas, 31.54% in serous borderline tumors, 18.86% in low-grade serous carcinomas, and 2.7% in HGSOCs (Figure 2A–D and Table 1). Among 39 mucinous tumors (9 benign, 12 borderline, 5 low-grade carcinomas, and 13 high-grade carcinomas), only one benign cystadenoma (11.11%) and one high-grade carcinoma (7.69%) had CAPS+ cells. In both cases, CAPS+ cells comprised approximately 20% of total epithelial cells (Appendix A). In endometrioid carcinomas, CAPS+ cells were present in 8 of 9 (88.88%) low-grade and 9 of 29 (31.03%) high-grade carcinomas. The average percentage of CAPS+ cells for all cases was markedly different between low-grade and high-grade endometrioid carcinomas (43.33% vs. 4.31%, Table 1). Regarding clear cell tumors, only high-grade carcinomas were available. CAPS+ cells were observed in 9 of 30 (30%) of cases and the average percentage of CAPS+ cells per case was 3.2% (Table 1). Overall, CAPS+ cells were observed in 45% of the serous, 45% of the endometrioid, and 30% of clear cell compared to only 6% of mucinous carcinomas (*p* = 0.01).

### 3.2. mRNA Expression of Ciliated Cell Markers Is Higher in Borderline Tumors and Low-Grade Carcinomas than in High-Grade Carcinomas

We determined the mRNA expression levels of ciliated cell markers (FOXJ1 and CAPS) and a secretory cell marker (PAX8) in the CSIOVDB database, which includes 2433 serous, 185 endometrioid, 146 clear cell, and 78 mucinous ovarian carcinomas, 106 serous borderline (LMP) tumors, and 11 mucinous borderline (LMP) tumors. Both FOXJ1 and CAPS expression levels differed markedly between serous borderline tumors and serous carcinomas. The mean FOXJ1 expression in borderline serous tumors was 7.568 compared to 6.867 in serous carcinomas (*p* = 2.35 × 10^−16^) (Figure 4A). Similarly, the mean CAPS expression in serous borderline serous tumors was 9.508 compared to 8.207 in serous carcinomas (*p* = 3.09 × 10^−27^). In contrast, the mean PAX8 expression levels were similar in serous borderline tumors (9.978) and serous carcinomas (9.989; *p* = 7.03 × 10^−1^). Of note, mucinous tumors expressed low levels of both ciliated cell (FOXJ1, CAPS) and secretory cell (PAX8) cell markers, possibly suggesting a different cell of origin for this histologic subtype. FOXJ1 and CAPS expression levels also decreased with increasing tumor grade while PAX8 levels increased with tumor grade (Figure 4B). Thus, the mRNA expression results are highly concordant with the CAPS immunostaining results in TMAs 1–4 (Table 1 and Appendix A).

### 3.3. Cells Expressing Ciliated Cell Markers Are Present in a Higher Percentage of Primary Tumors Than in Patient-Matched Metastases

Using the “HGSOC Progression” TMA, we compared the percentage of 42 patient-matched primary, synchronous metastatic, and metachronous/recurrent metastatic tumors for the presence of FOXJ1+ and/or CAPS+ cells. In this TMA, the primary and synchronous metastatic tumors were treatment naïve while the metachronous/recurrent metastases were obtained post-chemotherapy. FOXJ1+ and/or CAPS+ positive cells were identified in 22 of 42 (52%) primary tumors, 11 of 42 (26%) synchronous metastases, and 10 of 42 (24%) metachronous metastases. These percentages are consistent with 39.13% CAPS+ HGSOC in TMAs 1–4 (Appendix A), considering that HGSOC samples in TMAs 1–4 were not necessarily primary tumors. The number of positive cells per core varied in the “HGSOC Progression” TMA, but in most cases where metastases contained FOXJ1+ and/or CAPS+ cells [8 of 11 (73%) synchronous and 8 of 10 (80%) metachronous], fewer positive cells were seen in the metastases than in the patient-matched primary tumor (Figure 5). Only 2 of 24 (8%) cases had FOXJ1+ and/or CAPS+ cells present in synchronous metastases but not in primary tumors. In one of those cases, both synchronous and metachronous metastases contained FOXJ1+ and/or CAPS+ cells.

## 4. Discussion

The role of ciliated cells in ovarian epithelial carcinogenesis is poorly understood. Recent evidence suggests that ciliated cells might help clear the oxidative and genotoxic stress within the fallopian tube that is related to ovulation and the production of follicular fluid [4]. As terminally differentiated cells with low proliferative potential, ciliated cells themselves might be less prone to the accumulation of DNA damage [17]. The loss of ciliated cells is also associated with menopause and aging, which are known risk factors for EOC [20,21].

The presence of ciliated cells may underlie EOC development. In contrast to the well-described model of high-grade serous ovarian carcinogenesis-with progression of TP53 mutation to STIC lesions [3,6,7], the origin of other EOC histologic subtypes is less understood. Although both clear cell and endometrioid OC might arise from endometrial tissue via endometriosis, they can be of ciliated and secretory lineage, respectively [3,6,7,12,22]. Alternatively, an EOC could originate from a CIC, an invagination of ovarian surface mesothelium or displaced fallopian tube epithelium into the ovarian stroma and subsequent metaplasia of the cyst lining into Mullerian-type epithelium that can be a precursor for various carcinoma histologies [13,23]. The origin of serous borderline/LMP and serous low-grade carcinoma also remains relatively poorly understood. Kurman et al. proposed that mutation burden within CIC could determine low vs. high-grade disease [23] and Laury et al. suggested that serous borderline/LMP tumors might be linked to fallopian tube secretory cell outgrowths (SCOUTs) [24]. This sequence could also originate with ovarian cystadenoma as a principal precursor, and serous borderline/LMP tumor and low-grade carcinoma as eventual pathogenic variants [25]. Confirming the presence of ciliated cell markers in serous, endometrioid, and clear cell EOC supports the hypothesis that these histologic subtypes originate from precursor tissues containing ciliated cells, such as the fallopian tube, endometrium, and/or CIC.

We found that ciliated cells were rarely present in mucinous tumors, irrespective of lesion grade or malignant potential. Most mucinous tumors expressed relatively low mRNA levels of ciliated cell markers FOXJ1 and CAPS and there was a clear dichotomy in PAX8 mRNA expression when mucinous EOC were compared to serous, endometrioid, and clear cell EOC. The origin of mucinous EOC has been difficult to determine and is often confounded by possible confusion with metastases from other organs [26,27,28]. There may be a difference between subtypes of mucinous cancer such as endocervical or gastrointestinal which contribute to these findings, although these data were not available for our cases [29,30]. Our study adds to the scant literature on the relationship between ciliated cells and the origin of mucinous EOC in contrast to the origin of other EOC histologic subtypes [31].

Our results are consistent with a recent study that reported more numerous CAPS+ cells and higher levels of CAPS mRNA expression in low-grade serous EOC compared to HGSOC [32]. Additionally, we found that the presence of CAPS+ cells is not limited to low-grade serous EOC as we detected CAPS+ cells in approximately 85% of serous cystadenomas, borderline tumors, and low-grade carcinomas compare to <40% of HGSOC. Interestingly, the average percentage of CAPS+ cells for all cases decreased with increasing grade aggressiveness of serous lesions (~40% in cystadenomas, ~30% in borderline tumors, ~20% in low-grade carcinomas, and <3% in HGSOCs. This finding is consistent with previous associations between the loss of ciliated cell marker FOXJ1 and worsened prognosis [18]. Endometrioid EOC exhibited a similar phenomenon; CAPS+ cells were observed in 89% of low-grade endometrioid carcinomas compared to 31% of high-grade endometrioid carcinomas and the average percentage of CAPS+ in all cases was 10-times higher in low-grade compared to high-grade endometrioid carcinomas.

In HGSOC, we found an additional decline in the presence of ciliated cell marker-positive cells when primary tumors were compared to patient-matched metastases. Ciliated cell markers were identified in 52% of primary HGSOC compared to approximately 25% of patient-matched synchronous and metachronous/recurrent metastases. Little is known regarding either the mechanism(s) by which ciliated cells are lost during ovarian cancer progression or the possible active role ciliated cells play in preventing cancer initiation and progression. In a premenopausal fallopian tube, ciliated cells comprise up to 50% of the tubal epithelial cells and are typically in direct contact with, or in close proximity to, secretory cells, which are the presumptive precursor cell type for ovarian cancer. The mechanism by which ciliated cells might protect secretory cells from genotoxic stress is unclear. One possibility is that the beating cilia provide physical protection like an umbrella over the secretory cells. Another possibility is that the juxtacrine and paracrine signaling between secretory and ciliated cells is needed to maintain homeostasis of the healthy cells and/or eliminate damaged and transformed cells from the epithelial monolayer through cell competition and extrusion. The reduction in the number of ciliated cells in menopause might alter this homeostasis and lead to the accumulation of damaged secretory cells that are vulnerable to tumorigenic transformation. However, it is less likely that ciliated cells have a protective role during the later stages of cancer progression as a sparse distribution of ciliated cells in tumors would dilute any physical or molecular means of protection. Based on the information currently available, fallopian tube and endometrial ciliated cells might initially be passively incorporated into the tumor mass. As terminally differentiated cells, ciliated cells cannot transdifferentiate or divide and are gradually outstripped by molecular (genetic, epigenetic, transcriptomic) conditions that result in a preferential proliferation of cancerous secretory-type cells during tumor progression. As such, ciliated cells are likely to be passive remnants of normal tissue engulfed by the tumor. Although ciliated cells in tumors might not have a protective role in tumor progression, their presence in a tumor might be a useful marker of reduced tumor aggressiveness, which is consistent with the observed positive correlation between the levels of ciliated cell markers and favorable clinical outcomes [18]. Our findings also raise the possibility that cellular mechanisms involved in tumor migration and/or adaptation to a new milieu (e.g., metastatic site) at least in part are responsible for, or contribute to, the decreased number of ciliated cells in HGSOC metastases compared to primary HGSOC.

Limitations to our study include the lack of demographic and clinical parameters, such as race, use of oral contraception, and use of hormone-replacement therapy, which might affect the number of ciliated cells. Our study was primarily histologic and did not link patient demographics to clinical outcomes. Our data do not clarify whether or not the presence or number of ciliated cells is determined by pre-existent patient or tumor characteristics or whether or not the number of ciliated cells present itself is prognostically significant. Additionally, although one of the largest studies to date, our number of TMAs and samples were relatively few and may be underpowered to detect differences. However, our study is strengthened by its utilization of a microarray validation dataset and by its unique focus on exploring ciliated cell markers in multiple stages of ovarian cancer progression.

In conclusion, we found that cells expressing ciliated cell markers were present in each of the five major histologic subtypes of EOC, and that ciliated cell marker-positive cells were more likely to be found in borderline and low-grade serous tumors compared to HGSOCs, as well as in primary HGSOCs compared to patient-matched metastases. Ciliated cell marker-positive cells were also less likely to be found in ovarian tumors of mucinous compared to other histologic subtypes. These findings support a model in which ciliated cell containing tissues, such as the fallopian tube, endometrium, and/or CIC, are the origin for the major histologic subtypes of EOC, with mucinous EOC being the exception. Future studies should focus on the mechanism(s) by which ciliated cells are lost during carcinogenesis, cancer progression, and metastatic spread. If ciliated cells provide protection from malignant transformation, understanding the mechanism of loss of ciliated cells may also lead to targets for prevention or treatment.

## Figures and Tables

**Figure 1 cells-11-04009-f001:**
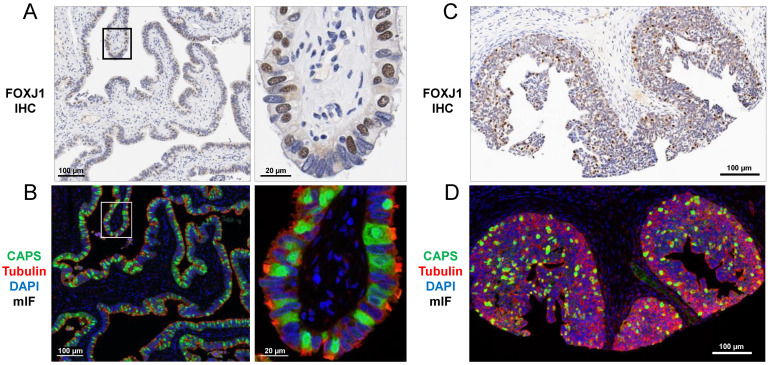
Ciliated cell markers FOXJ1 (IHC) and CAPS (mIF) in (**A**,**B**) normal fallopian tube and (**C**,**D**) proliferative endometrium. FOXJ1 shows nuclear staining (**A**,**C**) while CAPS shows diffuse intracellular staining (**B**,**D**). TUBB4 stains the cytoskeleton including base of cilia (**B**,**D**). Images (left) are shown at higher magnification (right). (**A**,**C**) Hematoxylin counterstain. (**B**,**D**) DAPI counterstain.

**Figure 2 cells-11-04009-f002:**
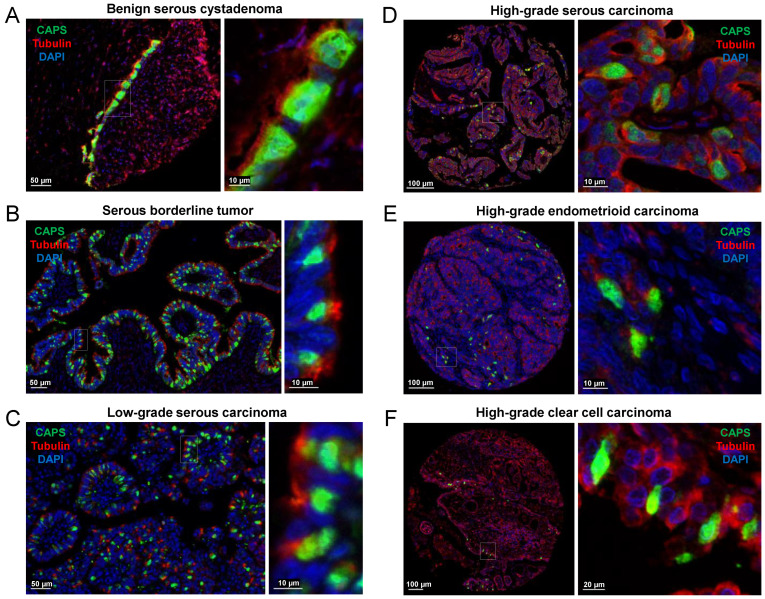
Ciliated cell markers CAPS and TUBB4 (mIF) in (**A**) benign serous cystadenoma, (**B**) serous borderline tumor, (**C**) serous low-grade carcinoma, (**D**) HGSOC, (**E**) high-grade endometrioid carcinoma, and (**F**) high-grade clear cell carcinoma. Images (left) are shown at higher magnification (right).

**Figure 3 cells-11-04009-f003:**
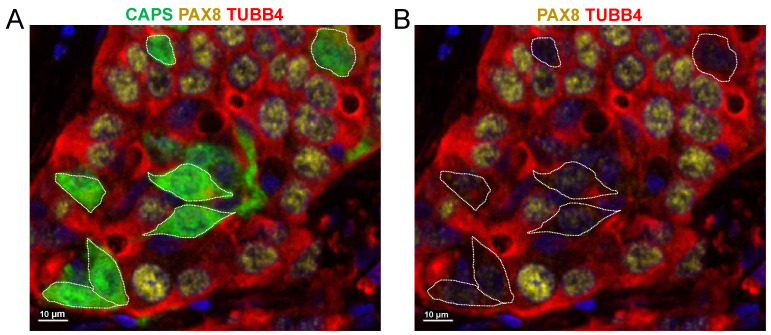
The nuclei of CAPS+ cells in HGSOC do not express PAX8. (**A**) CAPS, PAX8, and TUBB4 mIF, DAPI counterstain. (**B**) Same image as in (**A**) but without the CAPS channel (CAPS+ cells are outlined with a dotted line) for better visualization of the nuclear PAX8 staining.

**Figure 4 cells-11-04009-f004:**
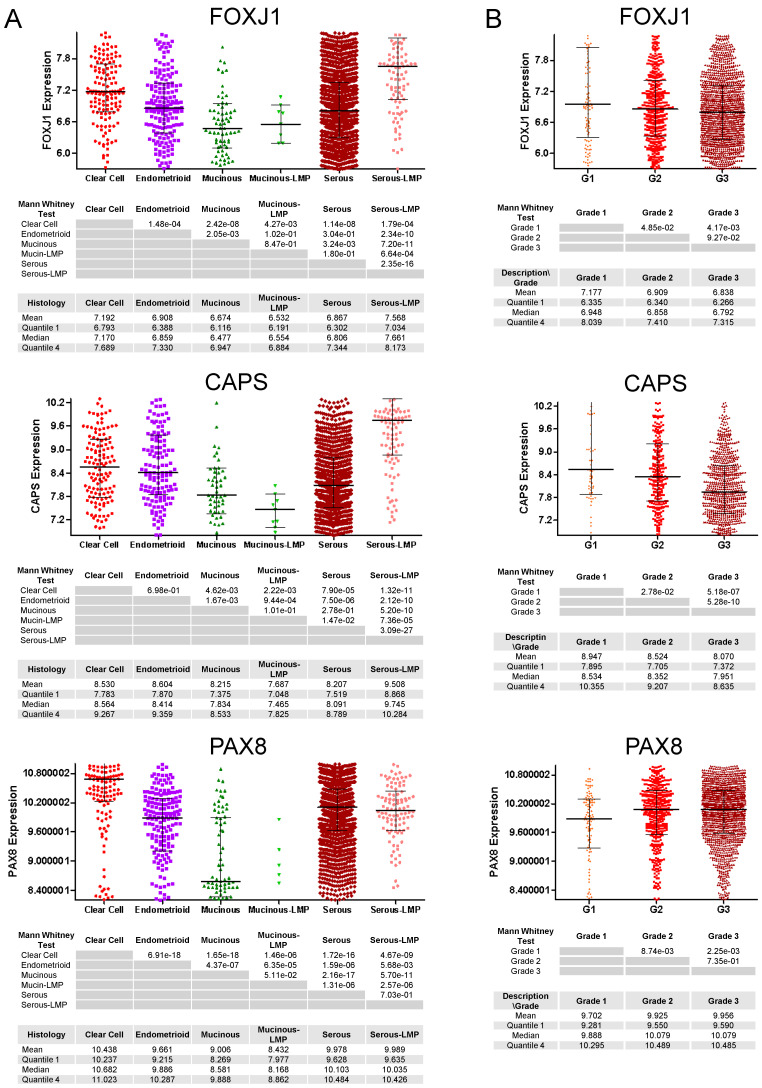
mRNA expression of ciliated cell markers FOXJ1 and CAPS and a secretory cell marker PAX8 in ovarian carcinomas (clear cell, endometrioid, mucinous, and serous) and borderline (LMP) tumors (mucinous, serous). (**A**) histologic subtypes and (**B**) cancer grade for all histologic subtypes combined. Y-axes represent relative mRNA levels.

**Figure 5 cells-11-04009-f005:**
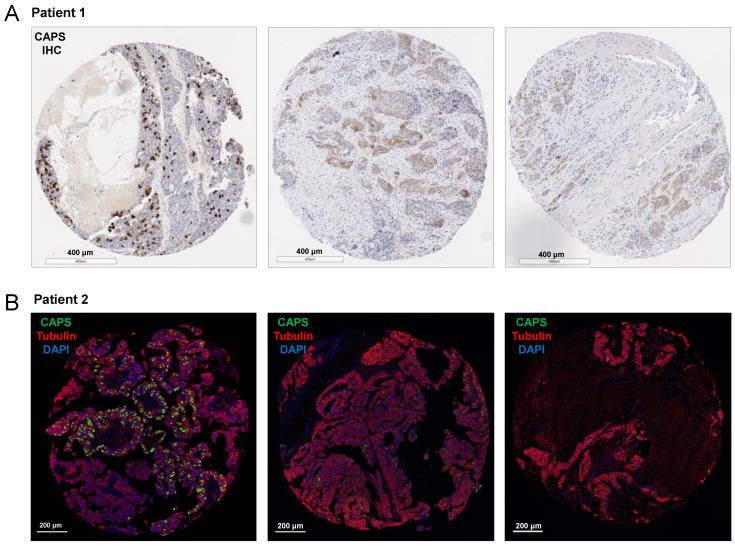
Detection of CAPS+ cells in primary, synchronous metastatic, and metachronous/recurrent metastatic HGSOC. Shown are two examples of HGSOC in which primary tumors exhibited high expression of CAPS by (**A**) IHC and (**B**) mIF.

**Table 1 cells-11-04009-t001:** Summary of CAPS immunostaining results in ovarian tumors.

**SEROUS**	**Ratio of**	**% CAPS+**	**Average % CAPS+ Cells**
	**CAPS+ Cases**	**Cases**	**in All Cases**
Benign cystadenoma	6/7	85.71	40.00
Borderline tumor	11/13	84.61	31.54
Low-grade carcinoma	6/7	85.71	18.86
High-grade carcinoma	18/46	39.13	2.70
**MUCINOUS**	**Ratio of**	**% CAPS+**	**Average % CAPS+ Cells**
	**CAPS+ Cases**	**Cases**	**in All Cases**
Benign cystadenoma	1/9	11.11	2.22
Borderline tumor	0/12	0.00	0.00
Low-grade carcinoma	0/5	0.00	0.00
High-grade carcinoma	1/13	7.69	1.53
**ENDOMETRIOID**	**Ratio of**	**% CAPS+**	**Average % CAPS+ Cells**
	**CAPS+ Cases**	**Cases**	**in All Cases**
Low-grade carcinoma	8/9	88.88	43.33
High-grade carcinoma	9/29	31.03	4.31
**CLEAR CELL**	**Ratio of**	**% CAPS+**	**Average % CAPS+ Cells**
	**CAPS+ Cases**	**Cases**	**in All Cases**
High-grade carcinoma	9/30	30.00	3.20

## Data Availability

The raw and processed data presented in this study are available upon request from the corresponding author.

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
