# Peer review of "Ciliated Cells in Ovarian Cancer Decrease with Increasing Tumor Grade and Disease Progression"

_cells, 2022, doi:10.3390/cells11244009_

Round 1

Reviewer 1 Report

The manuscript “Ciliated cells in ovarian cancer decrease with increasing tumor grade and disease progression” by Richardson MT et al., reported the loss of ciliated cells during the progression of ovarian cancer. The authors have used multiple cohorts of Ovarian Cancer TMAs to address the finding. The authors also have used the multiple markers of ciliated epithelium CAPS, FOXJ1 and TUBB4 and demonstrated by both IHC and multiplex IF. In addition, authors have analyzed mRNA expression of ciliated cell markers FOXJ1 and CAPS in CSIOVDB database and corroborated with the staining data. The findings are robust however, little incremental as it is already reported in ovarian cancer. Having a mechanistic insight how the loss of ciliated cells progress to ovarian cancer progression would be interesting. Using immortalized FTE cell lines and deleting ciliated cells and accessing the transformation and oncogenesis traits would establish its role as a tumor suppressor.

Author Response

We thank the reviewer for their overall positive review of our study.

We agree that our study confirms previous research into the loss of ciliated cells in low- and high-grade serous ovarian cancer, which has already been shown. We believe that our data are novel in that we have expanded the analysis to include:
a)      benign and borderline tumors
b)      other histologic subtypes (mucinous, endometrioid, and clear cell)
c)      patient-matched primary, metastatic, and recurrent ovarian cancers

d)      two different markers of ciliated cells (FOXJ1 and CAPS)

We also agree with the reviewer that it would be interesting to have a mechanistic insight into how the loss of ciliated cells might contribute to ovarian cancer initiation in the fallopian tube, and are working on this as one of the major projects in our laboratory, although these data are not yet mature. We have strong in vitro evidence that ciliated cells protect secretory cells from genotoxic stress and are currently generating transgenic mice that will allow us to validate these results in vivo.

With reference to the reviewer’s comments, we have added the following to our discussion section, page 11, lines 283-308: “In a premenopausal fallopian tube, ciliated cells comprise up to 50% of the tubal epithelial cells and are typically in direct contact with, or in close proximity to, secretory cells, which are the presumptive precursor cell type for ovarian cancer. The mechanism by which ciliated cells might protect secretory cells from genotoxic stress is unclear. One possibility is that the beating cilia provide physical protection like an umbrella over the secretory cells. Another possibility is that the juxtacrine and paracrine signaling between secretory and ciliated cells is needed to maintain homeostasis of the healthy cells and/or eliminate damaged and transformed cells from the epithelial monolayer through cell competition and extrusion. The reduction in the number of ciliated cells in menopause might alter this homeostasis and lead to the accumulation of damaged secretory cells that are vulnerable to tumorigenic transformation. However, it is less likely that ciliated cells have a protective role during the later stages of cancer progression as a sparse distribution of ciliated cells in tumors would dilute any physical or molecular means of protection. Based on the information currently available, fallopian tube and endometrial ciliated cells might initially be passively incorporated into the tumor mass. As terminally differentiated cells, ciliated cells cannot transdifferentiate or divide and are gradually outstripped by molecular (genetic, epigenetic, transcriptomic) conditions that result in a preferential proliferation of cancerous secretory-type cells during tumor progression. As such, ciliated cells are likely to be passive remnants of normal tissue engulfed by the tumor. Although ciliated cells in tumors might not have a protective role in tumor progression, their presence in a tumor might be a useful marker of reduced tumor aggressiveness, which is consistent with the observed positive correlation between the levels of ciliated cell markers and favorable clinical outcomes [18].

Reviewer 2 Report

The authors study the expression of ciliary markers (CAPS and FOXJ1) in epithelial ovarian cancer trying to understand the relationship between ciliated cell and the biological behavior of EOC. They perform immunohistochemistry analysis of commercial and inhouse TMAs. They also analyze by multiplex Immunofluorescence the expression of CAPS, TUBB4 and PAX8. To confirm the results, they carry out a bioinformatics study of a transcriptomic database.

The results are consistent and reported with excellent images.

However, the authors do not clarify whether the loss of expression of ciliary markers is due to dedifferentiation of the tumor cells during the malignancy process or is due to infiltration by tumor cells of other origin and replacement of the ciliated cells characteristic of healthy tissue. They should clarify these options.

Author Response

We thank the reviewer for their time and comments.

We agree that it is important to clarify if the loss of expression of ciliary markers is due to dedifferentiation vs. infiltration and replacement of ciliated cells by tumor cells. We have added the following to our discussion section, page 11, lines 283-308: “In a premenopausal fallopian tube, ciliated cells comprise up to 50% of the tubal epithelial cells and are typically in direct contact with, or in close proximity to, secretory cells, which are the presumptive precursor cell type for ovarian cancer. The mechanism by which ciliated cells might protect secretory cells from genotoxic stress is unclear. One possibility is that the beating cilia provide physical protection like an umbrella over the secretory cells. Another possibility is that the juxtacrine and paracrine signaling between secretory and ciliated cells is needed to maintain homeostasis of the healthy cells and/or eliminate damaged and transformed cells from the epithelial monolayer through cell competition and extrusion. The reduction in the number of ciliated cells in menopause might alter this homeostasis and lead to the accumulation of damaged secretory cells that are vulnerable to tumorigenic transformation. However, it is less likely that ciliated cells have a protective role during the later stages of cancer progression as a sparse distribution of ciliated cells in tumors would dilute any physical or molecular means of protection. Based on the information currently available, fallopian tube and endometrial ciliated cells might initially be passively incorporated into the tumor mass. As terminally differentiated cells, ciliated cells cannot transdifferentiate or divide and are gradually outstripped by molecular (genetic, epigenetic, transcriptomic) conditions that result in a preferential proliferation of cancerous secretory-type cells during tumor progression. As such, ciliated cells are likely to be passive remnants of normal tissue engulfed by the tumor. Although ciliated cells in tumors might not have a protective role in tumor progression, their presence in a tumor might be a useful marker of reduced tumor aggressiveness, which is consistent with the observed positive correlation between the levels of ciliated cell markers and favorable clinical outcomes [18].

We again thank the reviewers and editors for their comments to improve this manuscript.

Round 2

Reviewer 1 Report

Responses from the authors are satisfactory.